# Towards Mixed Reality Applications to Support Active and Lively Ageing

Marta Gabbi
258828@studenti.unimore.it
University of Modena and Reggio
Emilia
Reggio Emilia, Italy

Valeria Villani
valeria.villani@unimore.it
University of Modena and Reggio
Emilia
Reggio Emilia, Italy

Lorenzo Sabattini
lorenzo.sabattini@unimore.it
University of Modena and Reggio
Emilia
Reggio Emilia, Italy

## ABSTRACT

The global population is ageing at a significant pace. The percentage of older adults is expected to increase to 24% by 2030, leading to a consequent growth in the number of people affected by dementia. This condition calls for special attention to the daily physical and psychological needs of the individuals involved. Simultaneously, aged care problems, such as elevated costs and the shortage of professional caregivers, are worsening the scenario. Robotic technologies are playing a pivotal role in assisting elderly individuals in retaining their autonomy and providing innovative, direct assistance to them. In the literature, there is evidence of the significant impact that these technologies have on elderly people, especially the ones with dementia. These positive results translate into decreased depression and anxiety and into improved overall wellbeing and quality of life. This paper introduces a mixed reality application created for older adults with dementia, with the aim of stimulating motor and cognitive functioning. The application is deployed to Microsoft HoloLens 2 and the person is asked to select and play games with an increasing level of difficulty. This is intended as a first step that will pave the way towards effective and intuitive interaction with assistive robotic systems. The ultimate aim of this work is to help elderly people with dementia keeping their brain active and ageing in an environment filled with opportunities, rather than limitations.

## KEYWORDS

Mixed Reality, HoloLens, Dementia.

## 1 INTRODUCTION

The global population is ageing at a significant pace and the percentage of older adults is expected to increase to 24% by 2030. As a result of this growth, it is also increasing the number of individuals affected by dementia [1].

Dementia is a term that encompasses several diseases that affect the ability to perform daily activities. Dementia has physical, psychological, social and economic impacts not only on people diagnosed with it, but also on their families and their carers. Currently, more than 55 million people have been diagnosed with dementia worldwide, with Alzheimer disease being the most common form of dementia, contributing to 60-70% of cases. Every year, there are 10 million new cases, making this a crucial difficulty to face [2].

The growth in the ageing population is combined with a longer life expectancy, resulting in higher costs and challenges associated with providing care for the elderly. Among these challenges, we find the constant need for more professional caregivers and shrinking workforce.

Currently, there is no cure for dementia, but there are alternative forms of care that aim to improve the wellbeing of people with dementia by being physically active and involved in activities and social interactions that keep their mind engaged and maintain daily function.

Technology has proven to be an innovative resource to support patients affected by dementia, their families and their caregivers in many assisting tasks. Several studies have shown that interactions with human-like or animal-like robots can help improve the quality of life of elderly people with dementia. In the literature, there is also evidence of the effectiveness of virtual reality therapy for patients with dementia [3]. These positive results include a reduction in depressive symptoms, anxiety symptoms, and a decrease in negative physiological factors [4–6]. Simultaneously, an improvement in factors related to quality of life has been observed, along with an enhancement in interaction and communicative actions [7, 8].

Another important aspect of technology as an aid for people with dementia is the feeling of independence. These patients may feel like they are loosing their autonomy, having to rely on caregivers and family members for daily tasks. Technology can assist them during these activities, giving them the feeling of being independent.

Among the technologies that can be used, mixed reality (MR) can offer people with dementia cognitive stimulation and memory aid. MR merges virtual reality and augmented reality, creating an environment in which real and virtual objects and subjects coexist and interact. MR allows more immersive and dynamic interactions between the user and the holograms, integrated as part of the real world.

Given the importance of supporting people with dementia and helping them improve their overall well-being, this paper introduces a MR application, with the aim of enhancing short-term memory and visuo-spatial skills. We present a therapeutic activity in the form of multiple games, in which the user needs to collect objects and place them on the right surface. Our application is created using Unity as development platform and is deployed to Microsoft HoloLens 2, a wearable MR device that enables users to interact with holograms.

## 2 MIXED REALITY APPLICATION

This proposed MR application encourages movement in a known environment, helping to counteract the onset or progression of visuo-spatial disorders. Furthermore, it also stimulates cognitive functioning by requesting users to categorize the objects they collect.

Commonly, exercise with a therapist is carried out once a week or even less frequently, making it difficult to achieve continuity

of treatment. This application is designed for autonomous daily at-home exercise, facilitating consistent practice and engagement. The activities were thought and designed in collaboration with a neurologist, aiming to replicate exercises proposed by therapists in clinical settings.

## 2.1 Interaction Task

The aim of this activity is to stimulate the user's cognitive functioning and visuo-spatial abilities. At the beginning of the activity, the menu scene, which allows you to choose which game to play, is loaded using the user's head as origin of the spatial reference system.

As will be explained later in Section 2.2, the first thing the person needs to do is to change the origin of the coordinate system by scanning a QR code positioned in the room. The whole scene will then adjust its position with respect to the new origin. After this setup phase, the activity can start.

The first scene loaded is the menu scene. Thanks to the menu buttons, it is possible to choose which game to play among 3 different activities, designed with an increasing level of difficulty.

By selecting the first game, several red and green cubes will appear across the room, as shown in Fig. 1. The holograms are placed on the floor, on tables or other elements, blending in with the physical environment. The user is required to move within the space, collect the cubes and position them on the same-colored platform. Once the object is correctly placed, the counter located above the platform updates and a sound effect is produced. The game ends when all the cubes are placed on the right platform.

The second game is very similar but it is more challenging. In fact, the red and green cubes are not solid in color, but rather semi-transparent, making them harder to spot across the room. The environment is set up as shown in Fig. 2.

The third activity represents the highest level of difficulty and it is based on the Wisconsin Card Sorting Test (WCST) [9]. The WCST is a psychological test used in clinical and research settings to assess problem solving abilities, cognitive flexibility and abstract reasoning. A deck of cards is used, each containing different figures that vary in one or more characteristics, such as color, shape or number. The participant is asked to sort the cards according to different principles, without explicitly knowing the sorting rule. Moreover, the sorting rule changes periodically during the test. Similarly, in the game there are several objects that vary in shape and color, as shown in Fig. 3. There are four different gray platforms upon which the holograms have to be positioned. The participant is asked to move around the environment, find the objects and place them on the platform. Initially, the user is not aware of the correct sorting rule and has to deduce it by placing the objects on the platforms and observing when the score increases. After a set period of time, the sorting rule changes and the participant needs to guess the new one. The game ends when all of the objects are sorted correctly.

## 2.2 Development of the MR Environment

The application was developed using Unity, a real-time 3D development platform for building 2D and 3D applications, like games and simulations [10]. Unity uses Microsoft C# programming language and allows developers to create custom scripts for a game or an app.

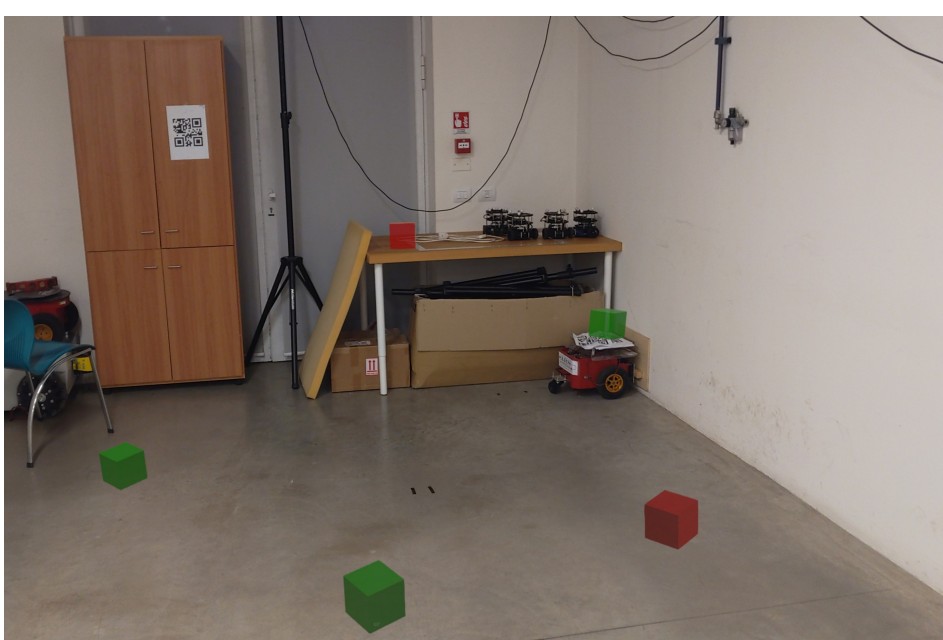

**Figure 1: First interaction task. Red and green cubes are spread across the room, blending in with the physical environment.**

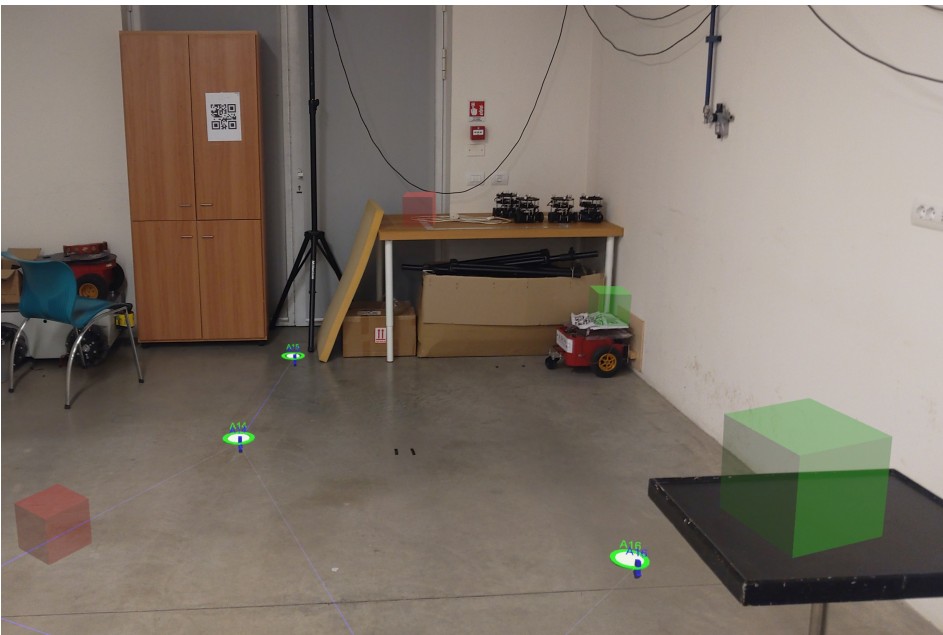

**Figure 2: Second interaction task. Red and green semi-transparent cubes are spread across the environment.**

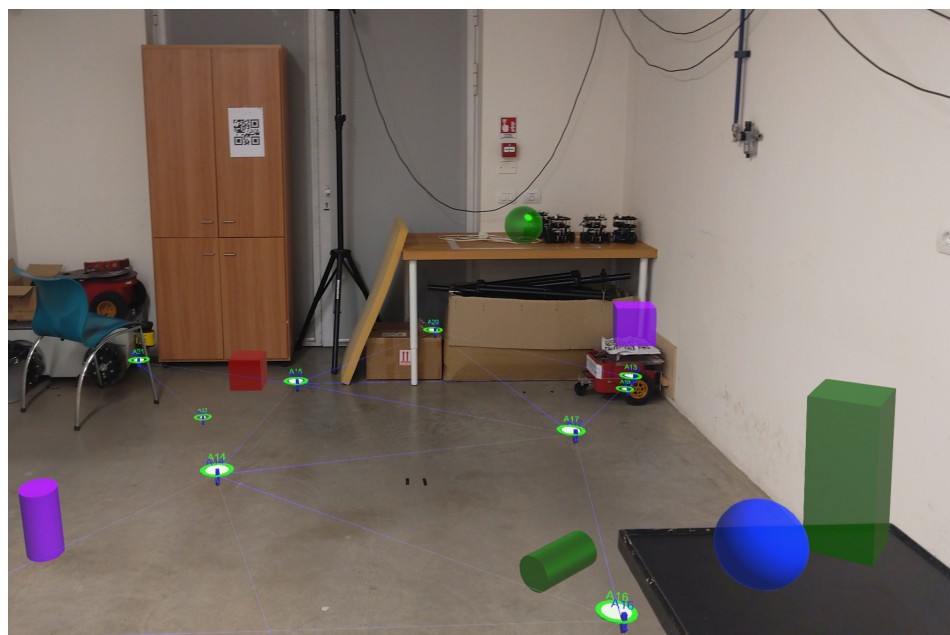

**Figure 3: Third interaction task. Several objects with different colors and shapes are found in the environment.**

For the generation of this application we used the Mixed Reality Toolkit (MRTK), which is an open-source toolkit developed by Microsoft for Unity [11]. It is used to accelerate the development of immersive cross-platform MR applications by providing a wide range of components, scripts and predefined resources that assist the developers. Among its common features, we can find gesture recognition, interaction with the real world and manipulation of virtual objects. MRTK also offers the capability to handle multiple scenes, thanks to the Scene System. One scene will be designated as manager scene, while the others will be content scenes. The application consists of four scenes: the menu scene, which is the manager, and one content scene for each game. The Scene System allows the application to work flawlessly, enabling the user to change scenes.

For the development of the app we also used the World Locking Tool (WLT), which provides a stable and reliable world-locked coordinate system [12]. It anchors the virtual world to the physical one by placing holograms in specific positions relative to real physical elements and other holographic objects. The implementation of this tool, particularly its *Space Pins* feature, has thus enabled the creation of an application that aligns the Unity space with the real world and addresses the scale error in holographic size perception on HoloLens applications.

Once the application is created in the Unity development environment, it was then deployed to HoloLens 2.

Microsoft HoloLens 2 is a wearable MR headset that enables users to interact with holograms integrated in the visualization of the real world [13]. It allows high-quality, high-fidelity, hands-free interaction. The MR headset has see-through holographic lenses that use a sophisticated optical projection system to generate multidimensional holograms with extremely low latency. HoloLens 2 is equipped with cameras and sensors, such as depth sensors, accelerometer and gyroscope, which are designed to capture information concerning what the user is doing and the surrounding environment.

An important functionality of HoloLens 2 is Spatial Mapping [14], which allows the device to create and update a real-time 3D map of the environment around the user. As the MR headset continuously collects new data about the environment and detects mutations within it, spatial surfaces will dynamically appear, disappear and change. This helps to better blend the holograms with the physical world, making it possible to place objects on real surfaces. This also means that the activity easily adapts to any environment, since it is mapped and recognized.

For HoloLens applications, the origin of the coordinate system is located at the center of rotation of the user's head, meaning that holograms are positioned in space relative to the user's head height. Although this ensures a more immersive experience, it also makes the application entirely tailored to the specific user. Thus, it is challenging to design an activity that is equally accessible to everyone.

To overcome this issue, we used the *Space Pins* feature of the WLT. We started with a sample project provided by Microsoft [15], which allows to recognize QR codes placed in the physical environment and to transform the coordinate system accordingly, setting the scanned point as the new origin. Building upon this project, we then developed the rest of our application, changing the structure of the scene and its components according to our requirements.

## 3 CONCLUSIONS AND FUTURE DEVELOPMENTS

The ageing of the population and the consequent increase in the number of elderly people affected by dementia is a long-term demographic trend that entails several challenges and problems. Technology can be an ally in addressing these challenges, supporting patients, families, and caregivers with a broad range of capabilities used in aged care.

The focus of this work is investigating the effectiveness of MR applications as a support for elderly people who might be subject to dementia, by designing activities aimed at maintaining and preserving cognitive functioning and visuo-spatial skills.

The next step of our research is to conduct a user study that both verifies the feasibility and the usability of our therapeutic MR application. This will be initially accomplished by introducing elderly subjects to Microsoft HoloLens 2 and seeing how they interact with the device. Afterwards, the user will test the application in a safe and known environment.

An important future step for our work would be to have elderly individuals use the application daily, aiming to observe if there are actual benefits resulting from the daily engagement in the activity.

As a further step, we aim at building upon this concept, to integrate the MR application with social robots: the main idea is that of leveraging on the ease of interaction and on the flexibility of application provided by MR technologies to pave the way towards extensive deployment of social robots to support elderly people in daily-life contexts.

## 4 ACKNOWLEDGEMENT

This work is supported by the "Lively Ageing" project, funded by the Italian Ministry of Health.

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
