# OpenReview forum: "Towards Mixed Reality Applications to Support Active and Lively Ageing"
_humanrobotinteraction.org/HRI/2024/Workshop/VAM-HRI — VAM-HRI 2024 Oral_

### Official Review · Reviewer_T2oH · 2024-02-23
**Accept**

**Rating:** 6
**Confidence:** 4

**Review:**

This paper provides a mixed reality prototype application to assist elderly populations. This is done by creating different mind games that aim to improve a user’s cognitive functioning and visuo-spatial abilities. Future work entails evaluating the effectiveness of this application in a user study, as well as integrating a social robot.

Strengths:

- This paper targets the elderly population, an important group of participants that are typically underrepresented.

- These applications clearly utilize the spatial aspects of mixed reality, making the HoloLens a suitable choice as the interaction device.

Areas of Improvement:

- The main area to improve is the formulation of how a robot might be incorporated into this work. Right now, it seems that building the application has been the main effort of this paper. Therefore, this would be an interesting paper to discuss at our workshop in the hopes of helping the authors devise a way to bring robots into their work!

---

### Official Review · Reviewer_Agmx · 2024-02-25
**Review B**

**Rating:** 6
**Confidence:** 5

**Review:**

The paper introduced MR application for elderly people with dementia to stimulate their cognitive functionality.

S1 Paper introduces interesting AR application for imporivng cognitive abilities of elderly people.

S2 Games/tasks are carefully designed.


W1 There is no releated work. I would encourage authors to review and add some articles doing similar work.

W2 Pilot studies would be quite useful.

W3 Task 3 could be better described.

Overall good work in progress paper!

---

### Decision · Program_Chairs · 2024-02-26

Accept (Oral)